# End-Stage Kidney Disease Resulting from Atypical Hemolytic Uremic Syndrome after Receiving AstraZeneca SARS-CoV-2 Vaccine: A Case Report

**DOI:** 10.3390/vaccines11030679

**Published:** 2023-03-16

**Authors:** Mohammed Tawhari, Moustafa S. Alhamadh, Abdulrahman Yousef Alhabeeb, Ziyad Almudayfir, Mansoor Radwi

**Affiliations:** 1College of Medicine, King Saud bin Abdulaziz University for Health Sciences (KSAU-HS), Ministry of the National Guard Health Affairs, Riyadh 14611, Saudi Arabia; alhamadhmo@gmail.com (M.S.A.); abdul.yh1@gmail.com (A.Y.A.); ziyad.almudaifer@gmail.com (Z.A.); 2King Abdullah International Medical Research Centre, Ministry of the National Guard Health Affairs, Riyadh 11481, Saudi Arabia; 3Department of Medicine, Division of Nephrology, King Abdulaziz Medical City, Ministry of the National Guard Health Affairs, Riyadh 11426, Saudi Arabia; 4Department of Hematology, College of Medicine, University of Jeddah, Jeddah 23218, Saudi Arabia; mradwi@uj.edu.sa

**Keywords:** atypical hemolytic uremic syndrome, aHUS, thrombotic microangiopathy, AstraZeneca vaccine, COVID-19 vaccine, SARS-CoV-2 vaccine

## Abstract

Hemolytic uremic syndrome (HUS) is classically described as a triad of nonimmune hemolytic anemia, thrombocytopenia, and acute kidney injury. Atypical HUS (aHUS) is a rare variant of the disease, and it accounts for 5–10% of the cases. It has a poor prognosis, with a mortality rate exceeding 25% and a more than 50% chance of progressing into end-stage kidney disease (ESKD). Genetic or acquired dysregulation of the alternative complement pathway is highly implicated in the pathogenesis of aHUS. Multiple triggers for aHUS have been described in the literature, including pregnancy, transplantation, vaccination, and viral infections. Herein, we report a case of a previously healthy 38-year-old male who developed microangiopathic hemolytic anemia and severe kidney impairment one week after receiving the first dose of AstraZeneca SARS-CoV-2 vaccine. A diagnosis of aHUS was made after excluding other causes of thrombotic microangiopathies. Treatment with plasma exchange, prednisone, and rituximab (375 mg/m^2^) once weekly for four doses resulted in improvement of his hematological parameters. However, he progressed to ESKD.

## 1. Introduction

Hemolytic uremic syndrome (HUS) is a thrombotic microangiopathic disease classically described as a triad of nonimmune hemolytic anemia, thrombocytopenia, and acute kidney injury (AKI) [1]. Anemia of HUS is severe in nature, with an elevated serum level of lactate dehydrogenase and fragmented red blood cells in peripheral blood smear [1,2]. HUS is mainly a disease of children, with more than 90% of the cases typically occurring secondary to infection with Shiga-like toxin producing Escherichia coli (O157:H7 serotype) or other bacteria, such as Streptococcus pneumoniae and Shigella [1,2]. Atypical hemolytic uremic syndrome (aHUS) accounts for 5–10% of HUS cases, and it is named so because it is not caused by the typical HUS triggers [3,4,5]. Genetic or acquired dysregulation of the alternative complement proteins, particularly factors H, I, and B; membrane cofactor protein; and complement 3, is implicated in the pathogenesis of aHUS in 40–60% of the cases [6]. aHUS has a poor outcomes, with a mortality rate exceeding 25% and a more than 50% chance of progression to end-stage kidney disease (ESKD) [3]. However, the advent of complement-target monoclonal antibodies, eculizumab, has attenuated the mortality and improved the renal outcomes. However, having prompt and definitive diagnosis is the key for successful treatment [7,8,9]. Many triggers for aHUS have been described in the literature, including pregnancy; transplantation; vaccination; and viral infections such as cytomegalovirus, Epstein–Barr virus, influenza, Varicella, and Hepatitis B, although no clear triggers are observed in 30% of aHUS cases [4,10,11,12]. Herein, we report a case of aHUS developed one a week following the administration of the first dose of AstraZeneca SARS-CoV-2 vaccine.

## 2. Case Presentation

A previously healthy 38-year-old male presented to our emergency department complaining of exertional shortness of breath, diffuse body weakness and fatigability, and swelling around both eyes one week after receiving the first dose of AstraZeneca SARS-CoV-2 vaccine. He denied a history of sick contact; fever; chills; cough; abdominal or flank pain; rectal bleeding; joint pain; nausea/vomiting; dysuria; and any changes in urine volume, color, and frequency. He did not have any comorbidities, and his family history was negative for kidney diseases and malignancies. He had no known allergy or a previous reaction to vaccines. On examination, the patient was not in distress and exhibited no signs of bleeding such as bruises or petechiae. He was vitally stable except for high blood pressure (169/70 mmHg). There was obvious orbital swelling with a puffy face and lower limb edema. The patient’s chest was clear with equal bilateral breath sound and normal vesicular breathing. His cardiovascular examination was unremarkable with normal S1 and S2 and no added sounds or murmurs. His abdomen was nontender, soft, and without evidence of organomegaly.

Initial laboratory investigations showed anemia (67 g/L) with ∼2.8% schistocytes on peripheral blood smear, thrombocytopenia (130 × 10^9^/L), kidney injury (creatinine: 960 umol/L, BUN: 43.8 mg/dL), evidence of non-immune hemolysis (low haptoglobin (<0.05), normal total and indirect bilirubin, elevated lactate dehydrogenase (658 U/L), and negative direct antiglobulin test), C-reactive protein (12 mg/L), erythrocyte sedimentation rate (36 mm/h), and ferritin (845 ug/L). Coagulation studies revealed a prothrombin time of 11.30 s, a partial thromboplastin time of 27.10 s, and an INR of 1.04 (Table 1). The polymerase chain reaction test for SARS-CoV-2 in nasopharyngeal swab was negative. Urinalysis showed cloudy urine with amorphous crystals (+3), hematuria (RBC: 18/HPF), and proteinuria (3.97 g/L). A urine culture was negative. A renal ultrasound was carried out and showed normal echogenicity, size, shape, and perfusion with preserved corticomedullary differentiation bilaterally. There was no evidence of calculi or hydronephrosis.

**Table 1 vaccines-11-00679-t001:** Important laboratory values at the time of admission and discharge.

Test Name:	Reference Range and Unit:	At Admission	At Discharge
Hgb	135–180 gm/L	67	85
MCV	76–96 fL	86.3	93.9
RBC	4.5–6.1 × 10^12^/L	2.24	2.64
Hct	0.42–0.54 L/L	0.194	0.248
RDW	11.5–14.5%	12.1	15.3
Retic percent	0.5–1.5%	4.81	2.34
Absolut retic	20.2–500 × 10^9^/L	135	64.30
WBC	4–11 × 10^9^/L	5.71	7.21
Neutrophils	2–7.5 × 10^9^/L	4.86	4.83
Lymphocytes	1–4.4 × 10^9^/L	0.67	1.44
Platelets	150–400 × 10^9^/L	130	162
PT	9.38–12.34 s	11.30	17.70
PTT	24.84–32.96 s	27.10	35.30
INR	0.80–1.20	1.04	1.67
Creatinine	64–110 umol/L	960	452
BUN	3.2–7.4 mmol/L	43.8	16.2
eGFR	60 mL/min/1.73 m^2^	6	13
AST	5–34 U/L	13	8
ALT	5–55 U/L	26	12
Alk Phos	40–150 U/L	38	48
Albumin	35–52 g/L	31	30
Uric acid	210–420 umol/L	601	284
LDH	125–220 U/L	658	515
CRP	8 mg/L	12	.
ESR	0–15 mm/h	36	.
Ferritin	21.8–274.6 ug/L	845	.
Sodium	136–145 mmol/L	132	133
Potasium	3.5–5.1 mmol/L	4.7	3.6
Calcium	2.1–2.55 mmol/L	1.97	2.32
Phosphorus	0.74–1.52 mmol/L	2.48	1.22

Abbreviations—Hgb: hemoglobin, MCV: mean corpuscular volume, RBC: red blood cell, Hct: hematocrit, RDW: red cell distribution width, WBC: white blood cell, PT: prothrombin time, PTT: partial thromboplastin time, INR: international normalized ratio, BUN: blood urea nitrogen, eGFR: estimated glomerular filtration rate, AST: aspartate aminotransferase, ALT: alanine aminotransferase, Alk Phos: alkaline phosphatase, LDH: lactate dehydrogenase, CRP: C reactive protein, and ESR: erythrocyte sedimentation Rate.

Thrombotic microangiopathy (TMA) was presumed given the classic triad of AKI, thrombocytopenia, and non-immune microangiopathic hemolytic anemia evidenced by negative antiglobulin test and schistocytes on peripheral blood smear. Due to the normal ADAMTS13 studies, thrombotic thrombocytopenic purpura was ruled out and aHUS was suspected. Further work ups, including autoimmune panel, complement protein levels, and gene panel for alternative pathway complement proteins, were ordered. The results of the autoimmune panel were negative for proteinase 3 and myeloperoxidase ANCA, the anti-glomerular basement membrane, and anti-nuclear and anti-dsDNA antibodies, and C3 and C4 complement levels were normal. Genetic work up came negative for C3; complement factor B (CFB); complement factor H (CFH); complement factor H-related protein 1; complement factor I, II, and V (CFI); complement regulatory protein CD46; thrombomodulin gene (THBD); complement receptor 1 (CR1); protectin (CD59); diacylglycerol kinase e (DGKE); inverted formin 2 (INF-2); methylmalonic aciduria cobalamin deficiency) chic type with homocystinuria (MMACHC); methylmalonyl-coa mutase (MMUT); phosphatidylinositol glycan biosynthesis class A protein (PIGA); and plasminogen (PLG). Stool culture was negative for Salmonella, Shigella, and Shiga-like toxin-producing Escherichia coli. In addition, heparin-induced thrombocytopenia (HIT) antibodies (PF-4 heparin complex) were negative. It should be noted that neither tests for autoantibodies to complement factors nor tests for the serum levels of different complement factors could be done as these tests are not available at our center. Given the inconclusive genetic, autoimmune, and complement work ups, a diagnosis of aHUS was made. Due to his severe renal impairment, the patient underwent hemodialysis (HD) through a tunneled catheter followed by kidney biopsy. Biopsy results were consistent with chronic kidney disease and features of TMA. There were glomerular fibrin thrombi with fragmented red blood cells in a background of moderate arteriosclerosis and hyalinosis, interstitial fibrosis, and tubular atrophy, and 14 out of 17 glomeruli were sclerosed. Electron microscope examination revealed endothelial thickening and cellular interposition of the glomerular basement membrane.

Initially, the patient was treated with plasma exchange while awaiting ADAMTS13 results. However, he developed left perinephric hematoma with evidence of active bleeding from the biopsy site. The bleeding was controlled successfully with angio-embolization and four units of fresh frozen plasma. Due to the increased bleeding risk and the normal ADAMTS13 studies, the plasma exchange was discontinued. Subsequently, the patient was commenced on prednisone (60 mg “1 mg/kg” per oral daily), rituximab (375 mg/m^2^) once weekly for four doses, and maintained on thrice weekly HD. After 4 weeks of treatment, he showed significant improvement in his blood count (Hgb: 85 mg/L with normal reticulocyte count, platelets: 162 × 10^9^/L, LDH: 515 U/L); however, he remained dialysis dependent. Figure 1, Figure 2 and Figure 3 illustrate the trajectories of hemoglobin, platelets, and creatinine. Following the rituximab treatment, his platelet count improved, and his hemoglobin stabilized. It should be noted that there was a drop in platelet count between weeks 4 and 5, yet the absence of schistocytes on peripheral blood smear and normal markers of hemolysis indicated that the drop was not related to aHUS activity. Most importantly, the platelet count did not decrease below the normal range. He was discharged with no active complaint on a tapering dose of prednisone, amlodipine, labetalol, calcium carbonate, and alfacalcidol, and he continued thrice weekly HD. On follow up 3 months later, he was doing well with stable blood count (Hgb: 88 mg/L, platelets: 224 × 10^9^/L, LDH: 277 U/L); however, he remained dialysis-dependent and was labeled as ESKD.

## 3. Discussion

Since the beginning of the pandemic, millions of vaccines have been administered worldwide, which has resulted in a significant decline in COVID-19 morbidity and mortality [10]. Although all of the approved SARS-CoV-2 vaccines have proven to be effective with tolerable local and systemic reactions such as pain, swelling, lymphadenopathy, headache, and myalgia, there are published case reports of unusual adverse effects such as myocarditis, thrombotic thrombocytopenia, and cerebral and splenic venous thrombosis [13,14,15,16]. Genetic or acquired dysregulation of the alternative complement pathway is highly implicated in the pathogenesis of aHUS [3,6]. In genetically susceptible individuals, many triggers have been linked to the development of aHUS, including pregnancy; transplantation; vaccination; and viral infections such as cytomegalovirus, Epstein–Barr virus, influenza, varicella, and Hepatitis B [4,10,11,12].

In this article, we call attention to a case of aHUS presenting with severe renal impairment and TMA developing a week after the first dose of the AstraZeneca SARS-CoV-2 vaccine. To the best of our knowledge, our case is unique as the patient did not have any detectable gene mutation associated with aHUS. Ferrer F et al. reported a case of aHUS associated with SARS-CoV-2 vaccine in a 54-year-old female with a previous history of pulmonary tuberculosis. Like our case, they reported negative pathogenic variants in CFH, CD46, CFI, C3, THBD, CFB, CFHR5, CFHR1, CFHR3, CFHR4, DGKE, and ADAMTS13, but they mentioned deletion of CFHR3/CFHR1, which could possibly explain the susceptibility for aHUS [17]. Similarly, Rysava R et al. reported a case of a young lady who developed aHUS following the second dose of the mRNA SARS-CoV-2 vaccine. In contrast with our report, that lady was found to have a genetic predisposition to aHUS [18]. In our patient, there was no identifiable genetic susceptibility for aHUS development, and he has previously received all recommended vaccinations without any complications or adverse reactions. The underlying mechanism by which the SARS-CoV-2 vaccine triggers the development of aHUS is unclear. One possible mechanism is the binding of SARS-CoV-2 spike protein to heparin sulphate leading to direct activation of the alternative pathway of complements (APC) [19]. The activation of APC leads to endothelial dysfunction resulting in the clinical manifestations of aHUS, including hemolysis, thrombocytopenia, and AKI. The absence of genetic predisposition in our patient may favor the direct activation of APC as the underlying mechanism. Additionally, our case is unique as it showed that aHUS in association with SARS-CoV-2 vaccine can result in ESKD.

In addition to dialysis, we used rituximab (375 mg/m^2^) once weekly for four doses to treat our patient’s condition. Despite the improvement in his hemolytic parameters and thrombocytopenia, he remained dialysis-dependent. Previous reports have shown that aHUS has more than a 50% chance of progressing into ESKD even with the proper interventions [3]. Additionally, previous reports have used eculizumab to treat aHUS that developed following the administration of SARS-CoV-2 vaccines [17,18]. Eculizumab is a monoclonal antibody against the C5 complement protein, resulting in the inhibition of all complement pathways. In 40–60% of aHUS cases, the dysregulation of the alternative complement pathway is the culprit, and therefore eculizumab would be the perfect treatment option to reverse the pathology [6]. The previous report by Rysava R et al. showed that plasma exchange had only a transient effect on the patient, while eculizumab resulted in a more sustained remission [18]. The use of eculizumab has been shown to improve the outcomes of aHUS significantly with a good safety profile [7,8]. Unlike our patient, the patient in that report had a possible genetic predisposition, which justified the use of eculizumab. Since no evidence of complement protein dysregulation was found and because the approval of eculizumab will take several weeks, we decided to go with rituximab instead. Rituximab resulted in a sustained improvement of all the hematological parameters; however, there was no improvement in the kidney function. There are potential explanations for the lack of improvement in the kidney function despite the hematological remission. First, our patient presented late (one week after the vaccine) and with more severe kidney dysfunction, in contrast with the Rysava R et al. report, where their patient presented within the first 24 h and had only mild kidney impairment that did not require hemodialysis. Second, our patient had a kidney biopsy that showed more advanced disease with more interstitial fibrosis and advanced glomerular sclerosis, in contrast with Rysava R et al., where none of the glomeruli was sclerosed. These observations call for high vigilance to recognize the disease as early as possible to have the best chance of recovery, particularly from a kidney standpoint. To give the patient the best possible chance of recovery, we emphasize the need for early and definitive diagnosis of such a possible adverse event from the vaccine. We also advocate for the early initiation of eculizumab.

The absence of genetic predisposition in our patients raised the possibility of vaccine induced thrombotic thrombocytopenia (VITT) as a potential diagnosis rather than aHUS. However, the negative essay for anti-PF-4 and the absence of thrombotic events made the diagnosis of VITT less likely [20,21]. We believe that our patient had aHUS and the AstraZeneca SARS-CoV-2 vaccine was the trigger, given the short period between vaccination administration and the development of the disease. There are several reasons to believe that the vaccine was the trigger for the development of aHUS. First, our patient had no previous history nor a family history of aHUS. Second, the temporal relationship between the administration of the vaccine and the development of aHUS raised the suspicion of the vaccine being the trigger. Third, previous reports that shared similarities with our case also supported our conclusion (Table 2). Fourth, despite the chronicity of kidney biopsy, the absence of other pathology than TMA together with active extra-renal TMA indicated that the kidney lesion was TMA-related and was likely triggered by the vaccine rather than being a pre-existing CKD from a different etiology. In addition, there are no studies to predict the onset of interstitial fibrosis following TMA, particularly since kidney biopsies are not done routinely in the setting of TMA. It is possible that with severe TMA leading to ischemia and endothelial injury, fibrosis can develop within a short period of time, and this has been reported previously [18]. With the absence of genetic predisposition, it is possible that the vaccine led to the direct activation of the APC, resulting in endothelial dysfunction and TMA. However, due to the unavailability of complement factor levels and antibodies to complement factors, we cannot be certain of the exact mechanism of injury.

It should be emphasized that we do not question the benefit of SARS-CoV-2 vaccination nor discourage patients from receiving the vaccines that have proven to protect from a potentially lethal infection [13]. We should also acknowledge that there are more cases of serious complications, including aHUS and thrombotic thrombocytopenia, from SARS-CoV-2 infection than from the vaccines [22]. However, we report this case to raise awareness among physicians of such complications since early recognition and optimal management can potentially prevent the progression to ESKD and improve the clinical outcomes. We also acknowledge that the temporal association between the vaccine and the development of aHUS does not confirm the causality.

**Table 2 vaccines-11-00679-t002:** Summary for the reported cases of TMA following SARS-CoV-2 vaccine.

Author	Age	Gender	Comorbidities	Clinical Setting	Genetic Predisposition	Treatment Used	Kidney Outcome
Claes et al. [23]	38	Female	No comorbidities	TMA developed one day following a booster dose of mRNA-1273 COVID-19 vaccine (Moderna)	Homozygous for the membrane cofactor protein risk haplotype (MCP)	Seven sessions of plasma exchanges and eculizumab	Recovered
Schmidt et al. [24]	60	Female	Not mentioned	TMA developed 2 weeks following the first dose of Pfizer BioNTech vaccine	Heterozygous for CFH-H3 haplotype resulting low factor H level	Plasma exchange	Recovered
Ferrer et al. [17]	54	Female	Treated pulmonary tuberculosis and 1 spontaneous abortion in the 1st trimester	TMA developed 5 days following ChAdOx1 nCoV-19 vaccine	Homozygous CFHR3/CFHR1 gene deletion	Corticosteroids, plasma exchange, and eculizumab	Recovered
Rysaya et al. [18]	21	Female	Resected ovarian teratoma; epilepsy; and allergy to ibuprofen, pethidine, penicillin, cefuroxime, and wasp stings	TMA developed one day following the second dose of mRNA vaccine (Comirnaty)	Heterozygous carrier of a pathogenic variation in CFH and heterozygous for risk haplotype of CD46 gene	Corticosteroids, plasma exchange and eculizumab	Recovered
Kotekoglu et al. [25]	24	Female	No comorbidities	TMA developed 2 days following the first dose of Pfizer-BioNTech vaccine	Not performed	Corticosteroids, IVIG, and eculizumab	Recovered
Thelma et al. [26]	43	Male	No comorbidities	TMA developed 3 days following the second dose of Astra Zeneca COVID-19 vaccine	Not performed	Corticosteroids	Recovered
Bouwmeester et al. [27]	Mean age: 39.2 (the youngest: 10 and the oldest: 58)	3 Females and 2 males	Family history of aHUS (N = 2). Hypertension (N = 1). Crohn’s disease (N = 1). Antiphospholipid syndrome (N = 1)	5 patients who developed new onset or relapsed of aHUS within 3 days following Astra Zeneca (N = 2) or Pfizer-BioNTech (N = 3) vaccines	All of the five patients had heterozygous C3 variant	Eculizumab	Recovered
Kaimori et al. [28] *	72	Female	Diffuse large B-cell lymphoma and hyperthyroidism	Diffuse and fatal TMA 2 days following the first dose of Pfizer-BioNTech vaccine	Not performed	N/A	N/A

* Autopsy study. Abbreviations—TMA: thrombotic microangiopathy, MCP: membrane cofactor protein, CFH: complement factor H, CFHR1: complement factor H-related 1, CFHR3: complement factor H-related 3, and N/A: not applicable.

## 4. Conclusions

Our case draws attention to a rare and challenging diagnosis that is aHUS, following the administration of the first dose of AstraZenca SARS-CoV-2 vaccine. Despite our thorough investigations and complex treatment, the patient progressed to ESKD. We believe that late presentation contributed, at least partially, to the severity and irreversibility of kidney damage. This case calls for measures to increase the physicians’ awareness and educate patients of this potential complication to allow for early detection to improve patient outcomes.

## Figures and Tables

**Figure 1 vaccines-11-00679-f001:**
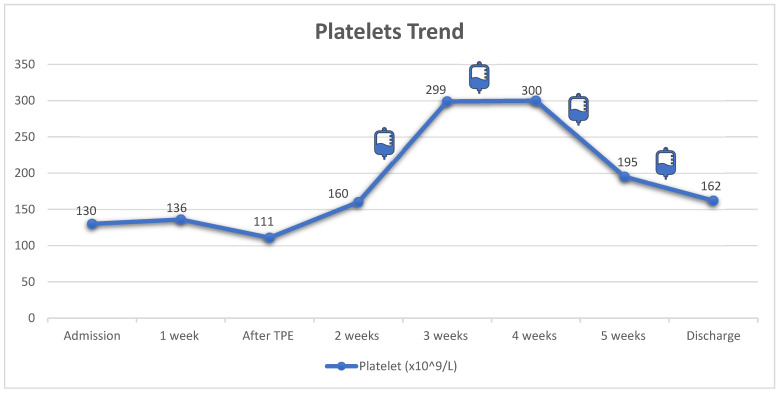
Trend of platelets count during hospital course. The infusion bag icons correlate with the time of each rituximab dose. Corticosteroids were not shown in the graph because the patient received daily dose since admission (IV methylprednisolone in the first 4 days and then switched to prednisolone tablets). TPE: therapeutic plasma exchange.

**Figure 2 vaccines-11-00679-f002:**
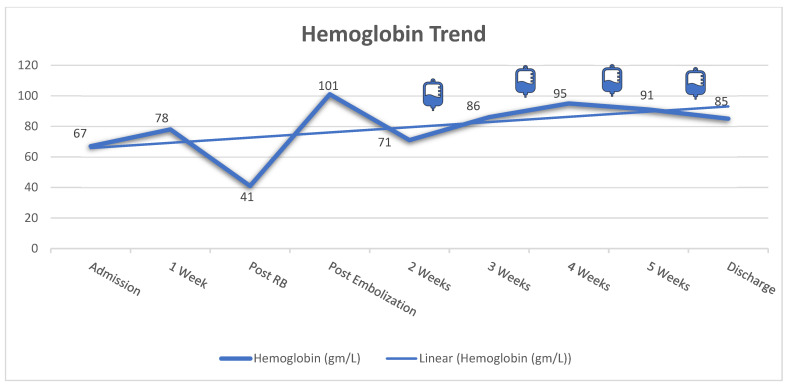
Trend of hemoglobin level during hospital course. The infusion bag icons correlate with the time of each rituximab dose. The patient had a significant drop in his hemoglobin level (41 gm/L) with perinephric hematoma post renal biopsy. Therefore, the left renal artery was successfully embolized, resulting in significant improvement in the patient’s hemoglobin level (101 gm/L). RB: renal biopsy.

**Figure 3 vaccines-11-00679-f003:**
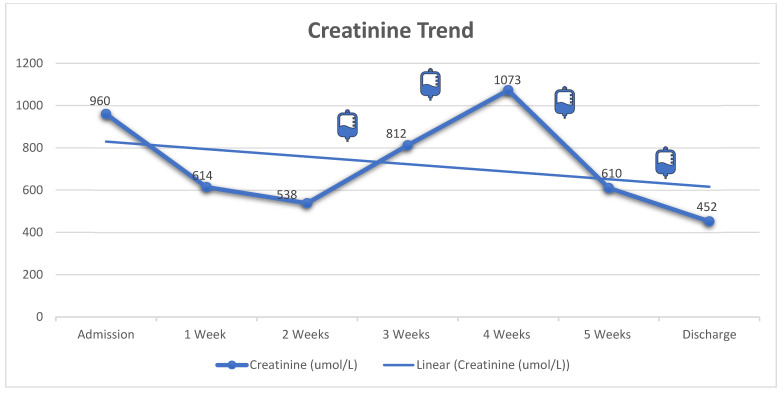
Trend of creatinine level during hospital course. The infusion bag icons correlate with the time of each rituximab dose. Rituximab resulted in a sustained improvement of all the hematological parameters (Figure 1 and Figure 2), but there was no improvement in kidney function.

## Data Availability

No new data were created or analyzed in this study. Data sharing is not applicable to this article.

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
