# Peer review of "End-Stage Kidney Disease Resulting from Atypical Hemolytic Uremic Syndrome after Receiving AstraZeneca SARS-CoV-2 Vaccine: A Case Report"

_vaccines, 2023, doi:10.3390/vaccines11030679_

Round 1

Reviewer 1 Report

This is a case report with no genetic abnormalities but a very high possibility of atypical hemolytic uremic syndrome (aHUS) after receiving first dose of AstraZenaca SARS-CoV-2 vaccinem who developed hemolysis that improved bud in end stage kidney failure (ESKD). I think your paper sis worth reporting as a case report of an adverse event involving a vaccine, but I have some question.

#From the abstract to the main text and the discussion, you wright that aHUS has a high mortality rate and ESKD rate, but the reference is in the 2009 NEJM. The paper builds on papers from 2004 and 2005. I think it would be better to cite more recent papers since the prognosis of aHUS has improved dramatically from introduction of eculizmab therapy. On top of that, I think you should be better to emphasize the need for prompt definitive diagnosis and treatment. 

#Abstract: Since this paper is about aHUS, I do not think it needs much explanation about typical HUS.

#Figer1 and 2: Figure 1 and 2 are combined into one figure, and it is easy to see when to include the time of plasma exchange and rituximab treatment.

#About etiological search; You were performing an autoantibody panel, complement protein assay, and genetic testing to alternative pathway for diagnosis aHUS. But the autoantibody panel did not seem to include antibody to Factor H or I. Also, in the complement protein assay, did you measure serum level of Factor H, I, or B?

#Therapy: Plasma exchange and Rituximab (RITU) have been used as treatment, but why did you use RITU? I do not think the reason is well written. You also mention that the reason eculizmab was not used was because no genetic abnormality was found, but if there is no genetic abnormality and antibodies to factor H or I are present, the alternative pathway is enhanced, so it may be effective enough. The effect of eculizmab is excellent, and it is easy to see that only just one dose of eculizmab improves hemolysis and increases platelets, so it seems that you should have administered it onece.

However, I think it would have been difficult to avoid renal failure due to very poor renal biopsy findings.

#Discussion: I think it would be more impactful to wright a summary regarding the features about the case first. (line 137)

I think it would be better to write a bit more about why you think Vac is the cause. For example, many things have been mentioned as triggers for aHUS, but how long ago was the most common prior occurrence etc.

Author Response

Thank you for your kind comments.

#From the abstract to the main text and the discussion, you wright that aHUS has a high mortality rate and ESKD rate, but the reference is in the 2009 NEJM. The paper builds on papers from 2004 and 2005. I think it would be better to cite more recent papers since the prognosis of aHUS has improved dramatically from introduction of eculizmab therapy. On top of that, I think you should be better to emphasize the need for prompt definitive diagnosis and treatment.

  • We added some updates in the introduction section and we cited more recent papers, and we acknowledged the role of eculizumab in prognosis. Also, we tried to emphasize the importance of having a definitive and timely diagnosis and, therefore, treatment with eculizumab.

#Abstract: Since this paper is about aHUS, I do not think it needs much explanation about typical HUS.

  • We deleted some information about typical HUS.

#Figer1 and 2: Figure 1 and 2 are combined into one figure, and it is easy to see when to include the time of plasma exchange and rituximab treatment.

  • We added icons showing the time of each Rituximab dose. Also, in figure 1, we added plasma exchange to the x-axis.

#About etiological search; You were performing an autoantibody panel, complement protein assay, and genetic testing to alternative pathway for diagnosis aHUS. But the autoantibody panel did not seem to include antibody to Factor H or I. Also, in the complement protein assay, did you measure serum level of Factor H, I, or B?

  • Unfrequently, we could not test for autoantibodies to complement factors nor for the serum level of different complement factors as these tests are not available at our center. We acknowledged this limitation in our paper.

#Therapy: Plasma exchange and Rituximab (RITU) have been used as treatment, but why did you use RITU? I do not think the reason is well written. You also mention that the reason eculizmab was not used was because no genetic abnormality was found, but if there is no genetic abnormality and antibodies to factor H or I are present, the alternative pathway is enhanced, so it may be effective enough. The effect of eculizmab is excellent, and it is easy to see that only just one dose of eculizmab improves hemolysis and increases platelets, so it seems that you should have administered it onece.

However, I think it would have been difficult to avoid renal failure due to very poor renal biopsy findings.

  • While waiting for the ADAMTS13, the patient was initiated on plasma exchange, however, he developed post-biopsy bleeding and we had to switch to Rituximab, particularly since the approval of eculizumab requires several weeks. Rituximab improved all patient’s hematological parameters, but unfortunately, there was no improvement in his kidney function. We believe that the patient kidney was not salvageable, given the advanced changes in the biopsy, so we did not switch to eculizumab.
  • We have made that clear to the reader. 

#Discussion: I think it would be more impactful to wright a summary regarding the features about the case first. (line 137)

I think it would be better to write a bit more about why you think Vac is the cause. For example, many things have been mentioned as triggers for aHUS, but how long ago was the most common prior occurrence etc.

  • Added as requested. Please see line 229-247.

Reviewer 2 Report

The paper is a Case report on a previously healthy 38-year-old male who developed microangiopathic hemolytic anemia and severe kidney impairment one week after receiving the first dose of AstraZeneca SARS-CoV-2 vaccine. The manuscript can be relevant for the field, but is not very clear and argumented. The following revisions should be made:

1.     Please provide more details on the introductory part;

2.     Line 122 – Figure 1. Please explain the figure in additional text included in the caption.

3.     Line 124 – Figure 2. Please explain the figure in additional text included in the caption.

The association of SARS-CoV-2 vaccine and aHUS is not very well demonstrated, as the patient also has kidney biopsy results reflecting “chronic kidney disease with features of 103 TMA, moderate arteriosclerosis and hyalinosis, moderate interstitial fibrosis, and tubular 104 atrophy with 14 out of 17 glomeruli were sclerosed” (lines 103-105). Please provide more information demonstrating your findings.  

Although I am not qualified to judge the English language, maybe the English proofreading will improve the overall quality of the paper.

Author Response

  1. Please provide more details on the introductory part;

- We have added a small part about eculizumab. The problem is that the number of words is limited to 2500. Because of that, we tried to shorten the introduction and focus on the discussion.

  1. Line 122 – Figure 1. Please explain the figure in the additional text included in the caption.

- Added. Please see figure 1-2.

  1. Line 124 – Figure 2. Please explain the figure in additional text included in the caption.

- Added. Please see figure 3.

The association of SARS-CoV-2 vaccine and aHUS is not very well demonstrated, as the patient also has kidney biopsy results reflecting “chronic kidney disease with features of 103 TMA, moderate arteriosclerosis and hyalinosis, moderate interstitial fibrosis, and tubular 104 atrophy with 14 out of 17 glomeruli were sclerosed” (lines 103-105). Please provide more information demonstrating your findings.  

  • The biopsy showed the following features: focal glomerular fibrin thrombi with fragmented RBCs in a background of moderate arteriosclerosis and hyalinosis, moderate interstitial fibrosis, and tubular atrophy with 14 out of 17 glomeruli were sclerosed. Examination by electron microscope showed endothelial thickening and cellular interposition of the glomerular basement membrane. All of which are features of TMA and CKD.
  • Despite the chronicity of kidney biopsy, the absence of other pathology than TMA together with active extra-renal TMA indicated that the kidney lesion was TMA-related and was likely triggered by the vaccine rather than being a pre-existing CKD from a different etiology
  • We clearly outlined the reasons that made us consider that the vaccine was the trigger for this presentation.

Although I am not qualified to judge the English language, maybe the English proofreading will improve the overall quality of the paper.

We have reviewed that paper and improved the language.

Reviewer 3 Report

This is an informative case report. Several concerns have arisen which should be addressed adequately.

1. Some speculation regarding the pathogenesis of the COVID-19 vaccine-induced aHUS should be added.

2. It is nice to use a table that summarized previously reported cases of the COVID-19 vaccine-induced aHUS and this case.

3. The authors used rituximab (RTX) for the treatment of the patient. Why the authors used RTX instead of eculizumab?

4. Discussion section should be concisely rewritten.

Author Response

Thank you for your kind review.

  1. Some speculation regarding the pathogenesis of the COVID-19 vaccine-induced aHUS should be added.

- Added. Line 187-194

  1. It is nice to use a table that summarized previously reported cases of the COVID-19 vaccine-induced aHUS and this case.

- Added. Please see table 2.

  1. The authors used rituximab (RTX) for the treatment of the patient. Why the authors used RTX instead of eculizumab?

- While waiting for the ADAMTS13, the patient was initiated on plasma exchange, however, he developed post-biopsy bleeding and we had to switch to Rituximab, particularly since the approval of eculizumab requires several weeks. Rituximab improved all patient’s hematological parameters, but unfortunately, there was no improvement in his kidney function. We believe that the patient kidney was not salvageable, given the advanced changes in the biopsy, so we did not switch to eculizumab.

  1. The discussion section should be concisely rewritten.

- We have made several changes to improve the discussion section.

Reviewer 4 Report

This is a very interesting case report by Tawhari et.al, addressing the association between developing aHUS after SARS-CoV-2 vaccine. Using hematology and genetic approach, identified aHUS in patient after receiving vaccine. Overall, presentation of case report data was good with table and figures and the conclusions made are well-founded with the presented results with discussion. However, there are MANY PLACES in the manuscript in which the presentation must be improved as noted below.

Line 75: Remove “consistent”

Line 80: Rewrite “Coagulation studies were unremarkable.”

Line 82 -83: What are the levels of hematuria and proteinuria?

Line 86: Table 1: Abbreviations for test names

Line 90: have you performed “schistocytes on peripheral blood smear” in this subject?

Line 96-97: Abbreviations for complement and others protein names.

Line 103-104: Histology of kidney was performed using PAS staining? The authors should present histology figures.

Line 111: dose of prednisone and what day it was started after admission?

Line 111: is it 4 doses or 4 weeks or 1 dose/week per 4 weeks?

Line 112: should be “4 weeks of treatment.”

Line 114: Units for platelet count. Consistent in writing units throughout the manuscript.

Line 115: Elaborate the results in figure 1 and 2

Line 122,124: add figure 1 and 2 legend along with y-axis label, and x-axis label should be 1 week after prednisone and rituximab

Line 121: Plot hemoglobin and platelet figures separately

Line 121-122: Why platelets were decreased suddenly between 4-5 weeks after admission?

Line 131: Should be “thrombosis.”

Line 157: Eculizumab blocks C5 cleavage results in inhibition of all three complement pathways

The discussion can also be improved and made more clear by at least one reference to each of the figures.

Author Response

Thank you for your kind review.

Line 75: Remove “consistent”

Modified into “Initial laboratory investigations showed anemia…” line 79

Line 80: Rewrite “Coagulation studies were unremarkable.”

Modified into “Coagulation studies revealed a PT of 11.30 seconds, a PTT of 27.10 seconds, and an INR of 1.04” line 84-85

Line 82 -83: What are the levels of hematuria and proteinuria?

Added. Line 87.

Line 86: Table 1: Abbreviations for test names

Added below table 1. Line 95-99

Line 90: have you performed “schistocytes on peripheral blood smear” in this subject?

Yes. Around 2.8% of schistocytes were found on the peripheral blood smear. line 79

Line 96-97: Abbreviations for complement and others protein names.

Added. Line 109-117.

Line 103-104: Histology of kidney was performed using PAS staining? The authors should present histology figures.

Unfortunately, the sample was sent out, and because of this, we could not get the histology pictures.

Line 111: dose of prednisone and what day it was started after admission?

The patient was on IV methylprednisolone for 4 days. Then, he was switched to oral prednisolone 60 mg “1mg/kg”. Line 132.

Line 111: is it 4 doses or 4 weeks or 1 dose/week per 4 weeks?

1 dose weekly for 4 weeks (4 cycles). We have made this clear in the paper.

Line 112: should be “4 weeks of treatment.”

Modified. Line 133.

Line 114: Units for platelet count. Consistent in writing units throughout the manuscript.

Fixed. The unit we used is x10^9/L and it is consistent.

Line 115: Elaborate the results in figure 1 and 2

Line 122,124: add figure 1 and 2 legend along with y-axis label, and x-axis label should be 1 week after prednisone and rituximab

Since corticosteroid was started since admission, it was not illustrated in the figures. We added icons showing the time of each Rituximab dose.

Line 121: Plot hemoglobin and platelet figures separately

Modified. Figure 1-2.

Line 121-122: Why platelets were decreased suddenly between 4-5 weeks after admission?

There was a drop in platelet count between weeks 4–5, yet the absence of schistocytes on peripheral blood smear and normal markers of hemolysis indicated that the drop was not related to aHUS activity. Most importantly, the platelet count did not decrease below the normal range. We have acknowledged that in the paper.

Line 131: Should be “thrombosis.”

Modified. Line 168

Line 157: Eculizumab blocks C5 cleavage results in inhibition of all three complement pathways

Modified. Line 200-201

The discussion can also be improved and made more clear by at least one reference to each of the figures.

We tried our best to improve the discussion.

Round 2

Reviewer 2 Report

The authors made the requested changes and added information providing sufficient background and description for the case report.

Reviewer 3 Report

The revised MS is almost addressed adequately.